# Comparative Genetic Characterization of Pathogenic *Escherichia coli* Isolated from Patients and Swine Suffering from Diarrhea in Korea

**DOI:** 10.3390/ani13071154

**Published:** 2023-03-24

**Authors:** Kyung-Hyo Do, Kwangwon Seo, Myunghwan Jung, Woo-Kon Lee, Wan-Kyu Lee

**Affiliations:** 1College of Veterinary Medicine, Chungbuk National University, Cheongju 28644, Republic of Korea; 2Department of Microbiology, College of Medicine, Gyeongsang National University, Jinju 52727, Republic of Korea; 3GutBiomeTech, Co., Ltd., Cheongju 28644, Republic of Korea

**Keywords:** swine, patients, diarrhea, *Escherichia coli*, virulence factors, antimicrobial resistance, multilocus sequence typing

## Abstract

**Simple Summary:**

The objective of this study was to compare the virulence and antimicrobial resistance characteristics of the most prevalent *Escherichia coli* strains causing diarrhea in both swine and humans. Results show that the swine strains exhibited a considerably higher level of resistance compared to those from human patients, particularly against fluoroquinolones. Moreover, five sequence types (ST 100, ST 1, ST 10, ST 641, and ST 88) were identified in both swine and human isolates. Furthermore, it was confirmed that both swine and human isolates possessed comparable virulence traits and shared similar phylogenetical relationships. These results suggest that cross contamination and the transmission of antimicrobial resistance may occur between swine and humans.

**Abstract:**

The aim of this study was to compare the virulence factors and antimicrobial resistance of the most common pathogenic *Escherichia coli* strains in swine and patients with diarrhea in Korea. We examined virulence genes and antimicrobial susceptibility in 85 and 61 *E. coli* strains isolated from swine and patients with diarrhea, respectively. The most prevalent pathogen in swine was enterotoxigenic *E. coli* (ETEC) (47.1%), followed by Shiga toxin-producing *E. coli* (STEC) (32.9%). Similarly, the majority of the patient isolates (50.8%) were proven to be STEC, the most common pathotype, followed by ETEC (23.0%). We found that swine isolates had significantly higher resistance than patient isolates, especially to fluoroquinolones (ciprofloxacin: 37.5% and 16.1%; norfloxacin: 29.7% and 16.1%, respectively). Additionally, sequence type (ST) 100 (swine: 21; patients: 4), ST 1 (swine: 21, patients: 2), ST 10 (swine: 8; patients: 6), ST 641 (swine: 3, patients: 2), and ST 88 (swine: 2, patients: 11) were detected in both swine and humans. In addition, we confirmed that isolates from swine and patients had similar virulence traits and were phylogenetically similar. According to these findings, swine and humans are susceptible to cross infection and the transfer of antimicrobial resistance.

## 1. Introduction

*Escherichia coli* encoding specific virulence factors can cause colibacillosis, which is a frequent disease in both swine and humans [1]. Because the morbidity and mortality rates are very high, colibacillosis in swine has a considerable economic impact on swine husbandry [2]. In addition, pathogenic *E. coli* causes hemorrhagic colitis and diarrhea in humans, as well as potentially fatal consequences, including hemolytic–uremic syndrome (HUS) [3].

To control and treat colibacillosis, antimicrobial drugs are routinely used [4]. In Korea, swine receive the majority of the antimicrobial agents sold (507 tons, 55%), followed by poultry (155 tons, 17%) and cattle (99 tons, 11%) [5]. As a result, isolates from swine have much greater levels of antimicrobial resistance than isolates from other animals [5]. Given that swine may spread antimicrobial resistance to humans, antimicrobial resistance surveillance is essential [6].

Generally, *E. coli* is a part of normal intestinal flora; however, *E. coli* encoding specific virulence factors (toxins and adhesin genes) is pathogenic [7]. Enterotoxigenic *E. coli* (ETEC) produces heat-labile (LT) and heat-stable (ST) toxins, with ST toxin further classified into STa and STb kinds, as well as enteroaggregative *E. coli* heat-stable enterotoxin 1 (EAST-1). Shiga toxin-producing *E. coli* (STEC), also known as verotoxin-producing *E. coli*, produces the systemic vascular damaging Shiga toxin (Stx) 2. [2,3]. To produce enterotoxins and cause diseases, pathogenic *E. coli* needs to attach to the intestines of pigs first. ETEC and STEC can also produce one or more of the following fimbriae: F4 (K88), F5 (K99), F6 (987P), F18, and F41 (F7) [2,3,6,7]. Fimbriae play a crucial function in intestinal mucosa and epithelial cell adhesion [2,3,6,7]. These virulence factors fluctuate in frequency over time and depend on the host [8,9]. Virulence genes such as enterotoxins (LT, STa, STb, and Stx) and/or adhesins are often found in both swine and human isolates. Swine serve as carriers for pathogenic *E. coli* transmission to humans via foods such as pork products, thereby causing human diseases [2]. It is necessary to examine the virulence traits of strains from various hosts and diseases to determine the risk of zoonotic infections [10].

Although numerous studies have concentrated on the antimicrobial resistance of *E. coli*, the majority of studies have focused on commensal *E. coli*. Little research has been conducted on the relationship between antibiotic resistance and *E. coli* virulence factors in swine and patients with diarrhea. Thus, we compared the virulence factors and antimicrobial resistance of the most common pathogenic *E. coli* strains in swine and patients with diarrhea in Korea. Knowledge of antimicrobial resistance and the distribution of virulence factors in *E. coli* is useful for designing treatment and prevention strategies and estimating the risk of human–swine cross infection.

## 2. Materials and Methods

### 2.1. E. coli Strains

According to a previous study, the 85 most prevalent pathogenic *E. coli* isolates from swine were employed in this study, especially those suffering from edema disease and post-weaning diarrhea [9]. Forty strains of ETEC, twenty-eight strains of STEC, and seventeen strains of ETEC/STEC were selected. For the isolation of these strains, intestinal and fecal samples were aseptically obtained and streaked on MacConkey agar (Becton Dickinson, Franklin Lakes, NJ, USA). Only pink colonies that appeared to be pure were identified as *E. coli*, and their identification was confirmed using a VITEK II system (bioMéreiux, Marcy I’Etoile, France). For further characterization, the examined isolates were stored in a 50% glycerol stock at −70 °C. One hundred and thirty-seven isolates from patients with diarrhea were provided by the National Culture Collection for Pathogens (NCCP, Republic of Korea), the Gyeongsang National University Hospital Branch of the NCCP (GNUH-NCCP, Republic of Korea), and the Kyungpook National University Hospital Branch of the NCCP (KNUH-NCCP, Republic of Korea).

### 2.2. Antimicrobial Susceptibility Test

According to the Clinical and Laboratory Standards Institute (CLSI) criteria, the following 20 antimicrobial agents were employed in this study: aminoglycosides (gentamicin, streptomycin, neomycin, kanamycin, and amikacin), a combination of beta-lactam/lactamase inhibitors (amoxicillin/clavulanic acid), 1st generation cephalosporins (cephalothin and cefazolin), 2nd generation cephalosporins (cefoxitin), 4th generation cephalosporins (cefepime), quinolones (nalidixic acid), fluoroquinolones (ciprofloxacin and norfloxacin), tetracyclines (tetracycline and doxycycline), aminopenicillin (ampicillin), folate pathway inhibitors (trimethoprim/sulfamethoxazole), phenols (chloramphenicol), polymyxins (colistin), and lincosamide (clindamycin) [11]. The antimicrobial discs used in this study were obtained from Becton-Dickinson. Antimicrobial susceptibility tests were performed according to CLSI guidelines and disk diffusion methods [12]. According to the Magiorakos criteria, isolates resistant to three or more antimicrobial subclasses were defined as multi-drug-resistant strains [13].

### 2.3. Multilocus Sequence Typing (MLST)

Macrogen (Seoul, Republic of Korea) was responsible for all procedures, including genomic deoxyribonucleic acid (DNA) extraction, polymerase chain reaction (PCR) amplification, Sanger sequencing, and assembly. InstaGene Matrix (Bio-Rad, Hercules, CA, USA) was used to extract the genomic DNA. The methodology for MLST involved the utilization of partial sequences from seven housekeeping genes (*adk*, *fumC*, *gyrB*, *icd*, *mdh*, *purA*, and *recA*), which had been previously described [8]. The primer sequences used for sequencing *adk*, *fumC*, *gyrB*, *icd*, *mdh, purA*, and *recA* are as follows: *adk* (forward: GCAATGCGTATCATTCTGCT; reverse: CAGATCAGCGCGAACTTCAG), *fumC* (forward: CCACCTCACTGATTCATGCG; reverse: CGGTGCACAGGTAATGACTG), *gyrB* (forward: CGGGTCACTGTAAAGAAATTAT; reverse: GTCCATGTAGGCGTTCAGGG), *icd* (forward: TACATTGAAGGTGATGGAATCG; reverse: GTCTTTAAACGCTCCTTCG G), *mdh* (forward: TCTGAGCCATATCCCTACTG; reverse: CGATAGATTTACGCTCTTC CA), *purA* (forward: CTGCTGTCTGAAGCATGTCC; reverse: CAGTTTAGTCAGGCAG AAGC), and *recA* (forward: AGCGTGAAGGTAAAACCTGTG; reverse: ACCTTTGTAGC TGTACCACG).

The PCR reactions were carried out with 20 ng of genomic DNA as the template in a 30 μL reaction volume using Dr. MAX DNA polymerase (MGMED, Inc., Seoul, Republic of Korea). The amplification protocol involved the activation of Taq polymerase at 95 °C for 5 min, followed by 35 cycles at 95 °C for 30 s, 52 °C for 30 s, and 72 °C for 1 min, with a final extension step of 10 min at 72 °C. After amplification, the resulting PCR products were purified using a multiscreen filter plate (Millipore Corp., Burlington, MA, USA). A PRISM BigDye Terminator v3.1 cycle sequencing kit (Thermo Fisher Scientific, Waltham, MA, USA) was employed for the sequencing procedure. After incubation at 95 °C for 5 min and a subsequent 5 min cooling on ice, the mixture was analyzed using an ABI PRISM 3730XL DNA analyzer (Applied Biosystems, Waltham, MA, USA). We assigned sequence types (ST) through an online platform (http://pubmlst.org/biqsdb?db=pubmlst_ecoli_achtman_seqdef, accessed on 12 September 2022).

### 2.4. Statistical Analysis

For the statistical analysis of the differences in antimicrobial resistance and virulence factors of *E. coli* between swine and patients, the chi-square test was performed using SPSS (version 12.0; SPSS Inc., Chicago, IL, USA).

## 3. Results

### 3.1. Pathotypes and Virotypes (Combination of Colonization Factors and Toxin Genes)

Table 1 shows a comparison of pathotypes and virotypes of *E. coli* from swine and patients (Table 1). The most prevalent pathotypes in the swine were ETEC (40 isolates, 47.1%) and STEC (28 isolates, 32.9%). However, the most prevalent pathotype in the patients was STEC (31 isolates, 50.8%), followed by ETEC (14 isolates, 23.0%). In addition, various pathotypes were confirmed in the patients: enteroaggregative *E. coli* (EAEC), enteroinvasive *E. coli* (EIEC), enteropathogenic *E. coli* (EPEC), STEC/EPEC, and STEC/EAEC. LT and ST toxin genes were found in ETEC isolates from both swine and patients. In STEC from patients, 23 isolates expressed the Stx1 gene, whereas Stx1 was not detected in any isolates from swine. In addition, STEC from patients had no detected colonization factors, whereas those from swine encoded the F18 and/or AIDA genes.

### 3.2. Antimicrobial Susceptibility Test

The results of antimicrobial susceptibility tests are described in Table 2. Resistance to gentamicin, neomycin, kanamycin, amikacin, nalidixic acid, ciprofloxacin, norfloxacin, tetracycline, doxycycline, trimethoprim/sulfamethoxazole, and chloramphenicol was significantly higher in swine isolates (51.8%, 51.8%, 89.4%, 75.3%, 70.6%, 41.2%, 32.9%, 87.1%, 80.0%, 56.5%, and 84.7%, respectively) than in the patient isolates (18.0%, 29.5%, 41.0%, 3.3%, 50.8%, 19.7%, 18.0%, 50.8%, 29.5%, 34.4%, and 34.4%, respectively). In contrast, amoxicillin/clavulanic acid (72.1%) resistance in patient isolates was significantly higher than that in swine isolates (36.5%). All isolates from swine were susceptible to cefepime, a fourth-generation cephalosporin, and two isolates (3.3%) from patients were resistant to cefepime.

### 3.3. Multidrug Resistance Rates

Figure 1 shows the results of the multidrug resistance analysis. Isolates from swine showed significantly lower non-multidrug resistance (0–2 antimicrobial resistance patterns; 4.7%) than isolates from patients (11.5%). In particular, 6–10 antimicrobial resistance patterns were significantly higher in isolates from swine (75.3%) than from patients (49.2%).

### 3.4. MLST

Figure 2 depicts a minimum spanning tree based on the MLST data, which includes branch distances. The number of divisions within each node was proportional to the number of ST-representing isolates. The black numerals in the circles indicate MLST sequence types. The red numbers on the line represent the absolute distances between the sequence types. Node diameters varied linearly with the number of ST-specific isolates. ST 1 (21 isolates, 31.8%) and ST 100 (21 isolates, 31.8%) were the most common STs found in the swine isolates. While the swine isolates included only 10 STs, the patient isolates contained 36 STs. ST 88 (11 isolates, 13.8%), ST 678 (6 isolates, 7.5%), and ST 10 (6 isolates, 7.5%) were the predominant STs among the patient isolates. Simultaneous detection of ST 100 (swine: 21 isolates; patients: 4 isolates), ST 1 (swine: 21 isolates, patients: 2 isolates), ST 10 (swine: 8 isolates; patients: 6 isolates), ST 641 (swine: 3 isolates, patients: 2 isolates), and ST 88 (swine: 2 isolates, patients: 11 isolates) occurred in both swine and patient samples. Furthermore, novel STs were found in five swine isolates, including ST New. In addition, ST 34–ST 218, ST 3744–ST 218, ST 10–ST 218, ST 90–ST 88, and ST New–ST 641 exhibited three absolute distances.

## 4. Discussion

In this study, we compared the virulence genes and antimicrobial resistance phenotypes of *E. coli* isolated from patients and swine with diarrhea. Colibacillosis symptoms differ based on the virulence factors encoded by the infective *E. coli* [14]. EPEC was reported as the most prevalent pathotype in patients in Korea in a previous study [15]; however, in the present study, we found that STEC was most prevalent, followed by ETEC. Until now, the reason for the temporal shifts in *E. coli* pathotypes has not been clarified; however, it is clear that the primary pathotypes were ETEC and STEC from EPEC. In this study, the majority of the swine isolates were confirmed to be STEC (28 isolates, 32.9%). Since STEC is an important zoonotic infection, careful treatment and prevention strategies for colibacillosis are needed [16]. Swine-to-human cross infection with STEC must be closely monitored because STEC can cause potentially life-threatening complications such as HUS [3,17].

Pathogenic *E. coli* must first attach to the intestine to create enterotoxins and cause illness [18]. Fimbriae are necessary for the first connection between the intestinal mucosa and epithelial cells [19]. F6 was the most common fimbriae in Korean swine in the late 1990s, but by the mid-2000s, it shifted to F5 [20,21]. However, no fimbrial adhesins were found in the patient isolates in this study, except for F4 and F18. F4 (29 isolates, 45.3%) and F18 (35 isolates, 54.7%) were the prevailing *E. coli* virotypes detected in the swine samples. Inactivated vaccinations targeting F4 and F18 have been utilized across Korea [22]. The use of these vaccines could result in antigenic variants and would explain why swine isolates also contain F4 and F18, in addition to fimbriae or non-fimbrial adhesins.

The Stx2 gene was found in both swine (45 isolates) and human (39 isolates) isolates. The Stx gene has been previously linked to edema in swine and HUS in humans [3,23,24]. The globotriosyl ceramide receptor for Stx2 is found in both swine and humans [25]. In addition, the LT and ST genes, which are known to be associated with diarrhea in swine and travelers’ diarrhea in humans, were found in both swine (LT: 42 isolates, ST: 43 isolates) and human isolates (LT: 11 isolates, ST: 13 isolates) [26]. Both enterotoxins raise extracellular chloride and bicarbonate levels, potentially causing osmotic diarrhea [27,28,29,30]. There were no common fimbrial adhesins in either swine or human isolates, except for F4 and F18. However, according to a recent study, LT may play an important role in improving bacterial adhesion. Thus, there is a possibility of cross infection with pathogenic *E. coli* between swine and patients [31].

Antimicrobial resistance was more common in swine isolates than in human isolates, which is in line with an earlier study [32,33]. This suggests that antimicrobial drugs are more frequently used in swine farming than in human medicine. Because antimicrobial use is not subject to tight control in Korea [34], non-specialists, such as livestock workers, may use antimicrobials indiscriminately, potentially increasing antimicrobial resistance. We found significantly higher resistance to several antimicrobials (aminoglycosides, quinolones, fluoroquinolones, tetracyclines, folate pathway inhibitors, and phenicols) in swine isolates than in patients. The highest rate of resistance was found in swine isolates, and resistance was found in drug classes approved for use in swine, indicating that pig antimicrobial usage may have contributed to the development of antimicrobial resistance in *E. coli* [35]. As these antimicrobial agents are used to treat human intestinal infections, lowering resistance to them is critical.

Fluoroquinolones are the most effective treatment for colibacillosis and have been routinely used in pig farms across the country [35,36]. As fluoroquinolones are useful in both human and animal medicine, they have been categorized as “critically important antimicrobial agents” by the World Health Organization and the World Organization for Animal Health [37]. A previous study found that Korean swine had high ciprofloxacin resistance (34.5%) [38], which is consistent with our findings. According to the results of this study, the resistance rate of swine isolates for ciprofloxacin was higher in Korea than in other advanced swine-farming countries (the Netherlands: 1.0%; Sweden: 0.0%; United States (US): 0.0%) where restrictions on the use of antimicrobials are tight [39,40]. This could be explained by the fact that fluoroquinolones are widely used for therapeutic and self-treatment purposes for livestock in Korea (quinolone sales: 44,380 kg). Furthermore, swine isolates had stronger resistance to ciprofloxacin (37.5% vs. 16.1% in patients) and norfloxacin (29.7% vs. 16.1%, respectively) than patient samples. Due to the possibility that fluoroquinolone resistance in swine could be transmitted to humans [41], it poses a public health risk, necessitating the development of an antimicrobial resistance reduction plan. Therefore, more control measures are required to limit ciprofloxacin resistance.

Aminoglycosides have been the principal choice for controlling colibacillosis since their introduction for clinical use in the mid-1990s, both in veterinary and human medicine [35,42]. In this study, there were higher rates of gentamicin resistance (51.8%) in Korea than in other advanced countries (US: 0.0%; Australia: 7.4%) [40,43]. Gentamicin is no longer used in the production of swine in advanced countries [43]; however, in Korea, gentamicin is still frequently used for the treatment of colibacillosis [5]. Korea has a high level of gentamicin resistance because of differences in antimicrobial agent use. Moreover, we found that swine isolates had significantly higher resistance to some aminoglycosides, including gentamicin, kanamycin, and amikacin, than patient isolates. Aminoglycosides are only used to treat severe infections in humans due to the risk of adverse effects such as sensorineural hearing loss and chronic kidney disease [44]. In contrast, aminoglycosides can be used to treat neonatal and post-weaning diarrhea in swine [44,45].

Treatment with aminoglycosides in swine can result in cross resistance to vital human antibiotics, such as amikacin, which is a serious health issue [43]. In this study, higher multidrug resistance (swine: 95.3%; patients: 88.8%) was observed in Korea than in other studies (in swine: the Netherlands: 34.2%, China: 84.2%, and Thailand: 84.6% [4,32,46]; and in humans: the Netherlands: 7.1%, China: 15.2%, and Thailand: 45.7% [14,46,47]). Compared to other Organization for Economic Co-operation and Development members, Korea uses antimicrobials in veterinary medicine at a rate of 33.2 defined daily doses (DDD) per 1000 people per day and in human medicine at a rate of 31.7 DDD per 1000 people per day (21.3 and 23.7 DDD per 1000 inhabitants per day, respectively) [48,49]. In addition to the potential impact of repeated exposure to therapeutic medicines leading to enhanced selective pressure, a cofactor such as the transfer of resistance genes and the spread of resistant isolates due to poor infection control could contribute to the rising prevalence of antimicrobial resistance among pathogens [16].

Antimicrobial resistance is closely linked to obstacles in treatment; thus, it is critical to manage antimicrobial resistance. In this study, we analyzed the virulence genes and antimicrobial resistance phenotypes of *E. coli* isolated from swine and patients suffering from diarrhea. The Stx2(e) gene, can cause serious diseases, including edema in swine and hemorrhagic colitis in humans, was found in both swine and human isolates. Antimicrobial resistance was much greater in swine isolates than in human isolates, particularly fluoroquinolones and aminoglycosides. These results could be valuable for the development of preventive and treatment strategies for enteric colibacillosis.

The utilization of MLST enables the establishment of phylogenetic connections among profound lineages, which presents an additional perspective on population structure [50]. The swine isolates examined in this investigation exhibited a limited number of STs (10 STs), whereas the patient isolates displayed a broader range of STs (36 STs). The primary STs found in swine were ST 1 (21 isolates, 31.8%) and ST 100 (21 isolates, 31.8%). This implies a comparable etiology of enteric colibacillosis in swine. The findings of this study are consistent with those of prior investigations that established ST 1 and ST 100 isolates as the dominant ETEC subtypes and identified them as significant swine pathogens in various countries, including the US, Canada, Germany, and Thailand (http://mlst.warwick.ac.uk/mlst/dbs/Ecoli, accessed on 12 September, 2022). In contrast to swine isolates, the STs of patient isolates had little overlap, indicating that each patient strain likely originated from a unique source. ST 10 has been identified as a commonly occurring strain in multiple studies involving human populations. This strain is frequently linked to the occurrence of distinct antimicrobial resistance genes, such as *ampC*-type beta-lactamases and NDM-type carbapenemases. [51,52]. A previous study linked ST 88 to c-*ampC* production in a French hospital [53]. Notably, both ST 10 (six swine isolates and three patient isolates) and ST 88 (two swine isolates and one patient isolate) were identified concurrently in both swine and patient samples. Furthermore, our results show that the isolates obtained from distinct hosts (swine and patients) were genetically related in the minimum spanning tree analysis. Moreover, we observed that five swine isolates with New ST were genetically similar to ST 641 in the phylogenetic analysis. While the link between patient isolates was weak, the emergence of a pathogenic *E. coli* strain with a novel sequence type represents a concerning issue not only for veterinary medicine but also for public health and therefore requires urgent attention. [54].

## 5. Conclusions

The presence of similar STs in swine and humans implies a potential risk of zoonotic disease emergence and cross-species transmission [55], as well as the possibility of transfer of antimicrobial resistance between the two populations in Korea.

## Figures and Tables

**Figure 1 animals-13-01154-f001:**
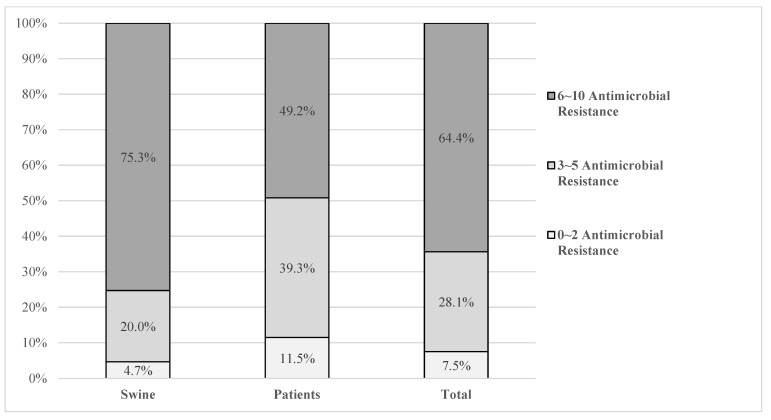
Multidrug resistance of *E. coli* from diarrheic pigs and patients in Korea. Antimicrobial subclasses defined by the Clinical and Laboratory Standards Institute (CLSI) are used.

**Figure 2 animals-13-01154-f002:**
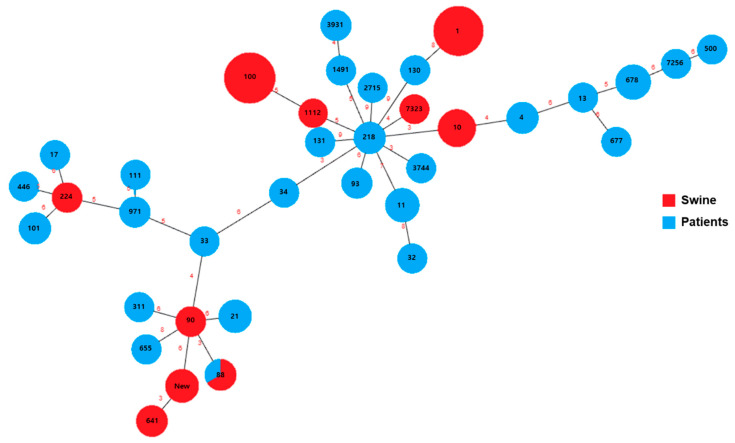
Minimum spanning tree based on sequence type of *E. coli* from pigs and patients. The divisions within each node are equal to the number of isolates belonging to the sequence type it represents. Black numbers in the circles indicate the MLST sequence type. Red numbers indicate the absolute distance between each sequence type. The node sizes vary linearly with the number of isolates of a given sequencing types.

**Table 1 animals-13-01154-t001:** Pathotypes and virotypes of *Escherichia coli* from diarrheic pigs and patients in Korea.

Pathotype /Virotype	No. (%) of Pathogenic *E. coli* Isolates			Chi-Square Value	Degrees of Freedom	*p*-Value
	Swine (*n* = 85)	Patients (*n* = 61)	Total (*n* = 146)			
**ETEC ***	**40 (47.1%)**	**14 (23.0%)**	**54 (37.0%)**	**12.695**	**1**	**0.000**
F4:LT:STb:EAST1 *	30 (75.0%)	-	30 (55.6%)	120.000	1	0.000
ST *	-	5 (35.7%)	5 (9.3%)	43.902	1	0.000
F4:paa:LT:STb:EAST1 *	1 (2.5%)	3 (21.4%)	4 (7.4%)	15.341	1	0.000
LT *	-	3 (21.4%)	3 (5.6%)	23.464	1	0.000
LT:ST *	-	3 (21.4%)	3 (5.6%)	23.464	1	0.000
F18:AIDA *	2 (5.0%)	-	2 (3.7%)	5.128	1	0.024
F4:F18:LT:STa:STb:EAST1 *	2 (5.0%)	-	2 (3.7%)	5.128	1	0.024
AIDA:STb:EAST1	1 (2.5%)	-	1 (1.9%)	3.046	1	0.081
F18:LT:STa:STb:EAST1	1 (2.5%)	-	1 (1.9%)	3.046	1	0.081
F18:paa:AIDA:STb:EAST1	1 (2.5%)	-	1 (1.9%)	3.046	1	0.081
F5:paa	1 (2.5%)	-	1 (1.9%)	3.046	1	0.081
**STEC ***	**28 (32.9%)**	**31 (50.8%)**	**59 (40.4%)**	**6.650**	**1**	**0.010**
F18:AIDA:stx 2e *	14 (50.0%)	-	14 (23.7%)	66.667	1	0.000
stx1:stx 2 *	-	12 (38.7%)	12 (20.3%)	48.447	1	0.000
stx1 *	-	11 (35.5%)	11 (18.6%)	43.902	1	0.000
stx2 *	-	8 (25.8%)	8 (13.6%)	29.885	1	0.000
F18:stx2:stx2e *	7 (25.0%)	-	7 (11.9%)	28.571	1	0.000
F18:stx2e *	4 (14.3%)	-	4 (6.8%)	15.054	1	0.000
F18:AIDA:stx2:stx2e *	3 (10.7%)	-	3 (5.1%)	11.640	1	0.001
**EAEC ***	**-**	**3 (4.9%)**	**3 (2.1%)**	**5.128**	**1**	**0.024**
aggR *	-	3 (100.0%)	3 (100.0%)	200.000	1	0.000
**EIEC**	**-**	**1 (1.6%)**	**1 (0.7%)**	**2.020**	**1**	**0.155**
ipaH *	-	1 (100.0%)	1 (100.0%)	200.000	1	0.000
**EPEC**	**-**	**1 (1.6%)**	**1 (0.7%)**	**2.020**	**1**	**0.155**
eae *	-	1 (100.0%)	1 (100.0%)	200.000	1	0.000
**ETEC/STEC ***	**17 (20.0%)**	**2 (3.3%)**	**19 (13.0%)**	**14.198**	**1**	**0.000**
F18:LT:STa:stx2e *	5 (29.4%)	-	5 (26.3%)	35.294	1	0.000
F18:st × 2:stx2e:EAST1 *	5 (29.4%)	-	5 (26.3%)	35.294	1	0.000
F18:LT:stx2e *	3 (17.6%)	-	3 (15.8%)	19.780	1	0.000
F18:paa:STa:stx2e *	2 (11.8%)	-	2 (10.5%)	12.766	1	0.000
F18:stx2e:EAST1 *	2 (11.8%)	-	2 (10.5%)	12.766	1	0.000
F18:paa:LT:STa:stx2:stx2e *	-	1 (50.0%)	1 (5.3%)	66.667	1	0.000
F18:paa:LT:STb:st × 2:stx2e:EAST1 *	-	1 (50.0%)	1 (5.3%)	66.667	1	0.000
**STEC/EPEC ***	**-**	**3 (4.9%)**	**3 (2.1%)**	**5.128**	**1**	**0.024**
stx1:eae *	-	3 (100.0%)	3 (100.0%)	200.000	1	0.000
**STEC/EAEC ***	**-**	**3 (4.9%)**	**3 (2.1%)**	**5.128**	**1**	**0.024**
stx2:aggR *	-	3 (100.0%)	3 (100.0%)	200.000	1	0.000
**Undetected ***	**-**	**3 (4.9%)**	**3 (2.1%)**	**5.128**	**1**	**0.000**

* Significant difference between origins of isolates (*p* < 0.05).

**Table 2 animals-13-01154-t002:** Antimicrobial resistance of *Escherichia coli* from diarrheic pigs and patients.

Antimicrobial Agent ^1^		No. (%) of Resistant Isolates			Chi-Square Value	Degrees of Freedom	*p*-Value
		Swine (*n* = 85)	Patients (*n* = 61)	Total (*n* = 146)			
**Aminoglycosides**	**GM ***	**44 (51.8%)**	**11 (18.0%)**	55 (37.7%)	25.407	1	0.000
	**S**	59 (69.4%)	36 (59.0%)	95 (65.1%)	2.170	1	0.104
	**N ***	44 (51.8%)	18 (29.5%)	62 (42.5%)	10.004	1	0.002
	**K ***	76 (89.4%)	25 (41.0%)	101 (69.2%)	50.637	1	0.000
	**AN ***	64 (75.3%)	2 (3.3%)	66 (45.2%)	108.953	1	0.000
**β-lactam/** **lactamase inhibitor**	**AMC ***	31 (36.5%)	44 (72.1%)	75 (51.4%)	24.700	1	0.000
**Cephalosporin 1**	**CF**	35 (41.2%)	26 (42.6%)	61 (41.8%)	0.082	1	0.774
	**CZ**	29 (34.1%)	16 (26.2%)	45 (30.8%)	1.524	1	0.217
**Cephalosporin 2**	**FOX**	25 (29.4%)	12 (19.7%)	37 (25.3%)	2.667	1	0.139
**Cephalosporin 4**	**FEP**	0 (0.0%)	2 (3.3%)	2 (1.4%)	3.046	1	0.081
**Quinolone**	**NA ***	60 (70.6%)	31 (50.8%)	91 (62.3%)	8.407	1	0.004
**Fluoroquinolone**	**CIP ***	35 (41.2%)	12 (19.7%)	47 (32.2%)	10.402	1	0.001
	**NOR ***	28 (32.9%)	11 (18.0%)	39 (26.7%)	5.922	1	0.015
**Tetracyclines**	**TE ***	74 (87.1%)	31 (50.8%)	105 (71.9%)	30.295	1	0.000
	**DOX ***	68 (80.0%)	18 (29.5%)	86 (58.9%)	50.505	1	0.000
**Aminopenicillin**	**AMP**	74 (87.1%)	47 (77.0%)	121 (82.9%)	3.388	1	0.066
**Folate pathway inhibitors**	**SXT ***	48 (56.5%)	21 (34.4%)	69 (47.3%)	10.666	1	0.001
**Phenicols**	**C ***	72 (84.7%)	21 (34.4%)	93 (63.7%)	53.968	1	0.000
**Polymyxins**	**CL**	1 (1.2%)	0 (0.0%)	1 (0.7%)	1.005	1	0.316
**Lincosamide**	**CC**	85 (100.0%)	61 (100.0%)	146 (100.0%)	Not available	Not available	Not available

^1^ GM: gentamicin; S: streptomycin; N: neomycin; CF: cephalothin; CZ: cefazolin; FEP: cefepime; FOX: cefoxitin; NA: nalidixic acid; CIP: ciprofloxacin; NOR: norfloxacin; AMP: ampicillin; AMC: amoxicillin/clavulanic acid; SXT: trimethoprim/sulfamethoxazole; C: chloramphenicol; CL: colistin; TE: tetracycline. * Significant difference between origins of isolates (*p* < 0.05).

## Data Availability

The datasets generated and/or analyzed during the current study are available from the corresponding author upon reasonable request.

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
