# Peer review of "Comparative Genetic Characterization of Pathogenic Escherichia coli Isolated from Patients and Swine Suffering from Diarrhea in Korea"

_animals, 2023, doi:10.3390/ani13071154_

Round 1
Reviewer 1 Report
In this manuscript, Do et al. submitted the research manuscript entitled “Comparative genetic characterization of pathogenic Escherichia coli isolated from patients and swine suffering from diarrhea in Korea”. The results showed the swine and humans are susceptible to cross-infection and the transfer of antimicrobial resistance. If the following problems are well-addressed, I believe that the essential contribution of this paper are important for reptile conservation.
1. In Simple Summary, I think that this section should not be a simple repetition of the abstract. Please rewrite this section according the requirements of the Animals.
2. In Methods, Lines 100-112, please provide primers for PCR reaction.
3. In Results, please provide relevant statistical results of chi-square test. These statistical values can be added in Table 2.
Specific comments:
Keywords: italicized “Escherichia coli”
Lines 68-72: Please rewrite this sentence to make it easier to follow
Author Response
Comment: In this manuscript, Do et al. submitted the research manuscript entitled “Comparative genetic characterization of pathogenic Escherichia coli isolated from patients and swine suffering from diarrhea in Korea”. The results showed the swine and humans are susceptible to cross-infection and the transfer of antimicrobial resistance. If the following problems are well-addressed, I believe that the essential contribution of this paper are important for reptile conservation.
Comment 1: In Simple Summary, I think that this section should not be a simple repetition of the abstract. Please rewrite this section according the requirements of the Animals.
Answer 1: As your suggestion, we revised “Simple Summary”.
Lines 12-20: The objective of this study was to compare the virulence and antimicrobial resistance character-istics of the most prevalent Escherichia coli strains causing diarrhea in both swine and humans. Results showed that the swine strains exhibited a considerably higher level of resistance com-pared to those from human patients, particularly against fluoroquinolones. Moreover, five se-quence types (ST 100, ST 1, ST 10, ST 641, and ST 88) were identified in both swine and human isolates. Furthermore, it was confirmed that both swine and human isolates possessed compara-ble virulence traits and shared similar phylogenetical relationships. These results suggest that cross-contamination and the transmission of antimicrobial resistance may occur between swine and humans.
Comment 2: In Methods, Lines 100-112, please provide primers for PCR reaction.
Answer 3: We added information about primers for PCR reaction on line 103 - 112 as your suggestion.
Lines 103 - 112: The primer sequences used for sequencing adk, fumC, gyrB, icd, mdh, purA, and recA are as follows: adk (forward: GCAATGCGTATCATTCTGCT; reverse: CAGATCAGCGCGAACTTCAG), fumC (forward: CCACCTCACTGATTCATGCG; reverse: CGGTGCACAGGTAATGACTG), gyrB (forward: CGGGTCACTGTAAAGAAATTAT; reverse: GTCCATGTAGGCGTTCAGGG), icd (forward: TACATTGAAGGTGATGGAATCG; reverse: GTCTTTAAACGCTCCTTCGG), mdh (forward: TCTGAGCCATATCCCTACTG; reverse: CGATAGATTTACGCTCTTCCA), purA (forward: CTGCTGTCTGAAGCATGTCC; reverse: CAGTTTAGTCAGGCAGAAGC), and recA (forward: AGCGTGAAGGTAAAAC-CTGTG; reverse: ACCTTTGTAGCTGTACCACG).
Comment 3: In Results, please provide relevant statistical results of chi-square test. These statistical values can be added in Table 2.
Answer 3: As your suggestion, we added the statistical values on Table 1 and Table 2.
Comment 4: Lines 68-72 Please rewrite this sentence to make it easier to follow
Answer 4: We revised lines 70-73 to reader friendly as your suggestion.
Lines 70-73: Forty strains of enterotoxigenic E. coli (ETEC), twenty-eight strains of Shiga tox-in-producing E. coli (STEC), and seventeen strains of ETEC/STEC were selected. For the isolation of these strains intestinal and fecal samples were aseptically obtained and streaked on MacConkey agar (Becton Dickinson, Franklin Lakes, NJ, USA).

Reviewer 2 Report
It was a pleasure to read the manuscript investigating pathogenic E. coli in both pigs and humans, and it shows some interesting results. It is generally well written and of interest to the readership
I have a few minor comments below, but nothing major. Where it is a grammatical issue, I have suggested the change
Line 50- virulence factors (toxins and adhesin genes) can be pathogenic [7].
I think that the gene names should be in italics throughout
Line 102- where previously described- is it possible to include a reference?
Line 104- could you include the PCR reagents?
Line 148-149- I think this is a bit muddled, it’s the isolate which shows resistance and not the antibiotic? Please check and reword if required
Line 184- is there a reason why this is ST New? Could it not be given a number?
Line 197- perhaps patients and swine with diarrhoea may sound better?
Line 216- were any of the pigs in the study vaccinated here?
Line 231- were the pigs treated with any antibiotics? Or is that not known? Also are growth promotor antibiotics used on farms? And if so, which one?
Line 266- Because of the risk of side. ….. (reword)
Line 279- 282- this is a bit vague- what cofactor? Perhaps needs a bit of explaining or clarification?
Line 315- delete full stop before reference 54
Author Response
Comment: It was a pleasure to read the manuscript investigating pathogenic E. coli in both pigs and humans, and it shows some interesting results. It is generally well written and of interest to the readership.
I have a few minor comments below, but nothing major. Where it is a grammatical issue, I have suggested the change
Comment 1: Line 50- virulence factors (toxins and adhesin genes) can be pathogenic [7]. I think that the gene names should be in italics throughout
Answer 1: Thank you for this suggestion. However, in this case, the LT, ST, and Stx is the abbreviation of toxin (protein).
So, I think this should be write in correct letter, not in italics.
Comment 2: Line 102- where previously described- is it possible to include a reference?
Answer 2: We added reference on line 103 as your suggestion.
Line 103: The methodology for MLST involved the utilization of partial sequences from seven housekeeping genes (adk, fumC, gyrB, icd, mdh, purA, and recA), which had been previously described [8].
Comment 3: Line 104- could you include the PCR reagents?
Answer 3: We already included the PCR reagents on line 98 - 119. These are the reagents for PCR and MLST.
- Genomic DNA extraction: InstaGene Matrix (BIO-RAD, cat.no.:732-6030)
- Taq pol.: Dr. MAX DNA Polymerase (Doctor protein INC, Korea, cat.no.:DR00302)
- PCR product purification: Multiscreen filter plate (Millipore Corp.)
- Sequencing Kit: BigDye(R) Terminator v3.1 Cycle Sequencing Kit (Applied Biosystems)
- Sequencer: ABI PRISM 3730XL Analyzer (96 capillary type)
Comment 4: Line 148-149- I think this is a bit muddled, it’s the isolate which shows resistance and not the antibiotic? Please check and reword if required
Answer 4: We revised line 157 - 159 as your suggestion.
Line 157-159: All isolates from swine were susceptible to cefepime, a 4th generation cephalosprins otherwise, two isolates (3.3%) from patients were resistant to cefepime.
Comment 5: Line 184- is there a reason why this is ST New? Could it not be given a number?
Answer 5: Yes. As you said, ST New is the new variant which is not be given a number.
Comment 6: Line 197- perhaps patients and swine with diarrhoea may sound better?
Answer 6: As your suggestion, we revised line 229.
Line 229: E. coli isolated from patients and swine with diarrhea.
Comment 7: Line 216- were any of the pigs in the study vaccinated here?
Answer 7: Unfortunately, we don’t know the exact information about vaccination of the pigs in this study. However, In Korea, inactivated vaccines for sows have been used countrywide. These vaccines contain E. coli whole cells with F4 and F18. So, we assumed the pigs in this study were vaccinated.
Comment 8: Line 231- were the pigs treated with any antibiotics? Or is that not known? Also are growth promotor antibiotics used on farms? And if so, which one?
Answer 8: Actually, we don’t know the exact information about recording of treatment for antimicrobial agents. However, as we mentioned, the regulation on the use of antimicrobials is not strict in Korea compared to other developed countries. So, we assumed these swine farms used growth promoting antimicrobials frequently.
Comment 9: Line 266- Because of the risk of side. ….. (reword)
Answer 9: As your suggestion, we revised line 299-300.
Line 299-300: Aminoglycosides are only used to treat severe infections in humans due to the risk of ad-verse effects such as sensorineural hearing loss and chronic kidney disease [44].
Comment 10: Line 279- 282- this is a bit vague- what cofactor? Perhaps needs a bit of explaining or clarification?
Answer 10: For reader friendly, we revised line 312-316.
Line 312-316: In addition to the potential impact of repeated exposure to therapeutic medicines leading to enhanced selective pressure, a cofactor such as the transfer of resistance genes and the spread of resistant isolates due to poor infection control could contribute to the rising prevalence of antimicrobial resistance among pathogens.
Comment 11: Line 315- delete full stop before reference 54
Answer 11: As your suggestion, lines were deleted full before reference 54.

Round 2
Reviewer 1 Report
The complete statistical parameters should include chi-square values, degrees of freedom, and P-values.
Author Response
As your suggestion, we added chi-square values, degrees of freedom on Table 1 and Table 2.